# Characterization of Spray-Dried Microcapsules of Paprika Oleoresin Induced by Ultrasound and High-Pressure Homogenization: Physicochemical Properties and Storage Stability

**DOI:** 10.3390/molecules28207075

**Published:** 2023-10-13

**Authors:** Qionglian Zhang, Yan Chen, Fang Geng, Xiaoyun Shen

**Affiliations:** 1School of Life Sciences and Engineering, Southwest University of Science and Technology, Mianyang 621010, China; zql20040912zql@126.com; 2School of Food and Biological Engineering, Chengdu University, Chengdu 610106, China; chenyancy@stu.cdu.edu.cn (Y.C.); gengfang@cdu.edu.cn (F.G.)

**Keywords:** paprika oleoresin, ultrasound, high-pressure homogenization, spray drying, storage stability

## Abstract

As an indispensable process in the microencapsulation of active substances, emulsion preparation has a significant impact on microencapsulated products. In this study, five primary emulsions of paprika oleoresin (PO, the natural colourant extracted from the fruit peel of *Capsicum annuum* L.) with different particle sizes (255–901.7 nm) were prepared using three industrialized pulverization-inducing techniques (stirring, ultrasound induction, and high-pressure homogenization). Subsequently, the PO emulsion was microencapsulated via spray drying. The effects of the different induction methods on the physicochemical properties, digestive behaviour, antioxidant activity, and storage stability of PO microencapsulated powder were investigated. The results showed that ultrasound and high-pressure homogenization induction could improve the encapsulation efficiency, solubility, and rehydration capacity of the microcapsules. In vitro digestion studies showed that ultrasound and high-pressure homogenization induction significantly increased the apparent solubility and dissolution of the microcapsules. High-pressure homogenization induction significantly improved the antioxidant capacity of the microcapsules, while high-intensity ultrasound (600 W) induction slowed down the degradation of the microcapsule fats and oils under short-term UV and long-term natural light exposure. Our study showed that ultrasound and high-pressure homogenization equipment could successfully be used to prepare emulsions containing nanoscale capsicum oil resin particles, improve their functional properties, and enhance the oral bioavailability of this bioactive product.

## 1. Introduction

Paprika oleoresin (PO) is a tetraterpenoid natural pigment extracted from chilli peppers as a deep red viscous oily liquid. PO contains a large number of carotenoids, such as capsanthin, capsorubin, β-carotene, zeaxanthin, cryptoxanthin, and lutein, as well as phenolic compounds, which have a high antioxidant capacity [1,2]. Studies have shown that PO can regulate the body’s lipid metabolism and immunity and has anti-radiation properties, as well as reducing the incidence of certain cardiovascular diseases [3]. Despite PO’s potential for application, these active compounds can be impractical to use due to their high viscosity and low water solubility, making them highly susceptible to the effects of temperature, light, and oxygen. However, this limitation can be overcome by encapsulation methods that slow down the rate of degradation of these compounds.

Microencapsulation utilizes certain stabilized materials to microencapsulate a core ingredient (e.g., paprika oleoresin). This is an effective way to enhance the stability of an unstable bioactive ingredient by providing a protective outer wall [4]. Among the various microencapsulation techniques that have been developed, spray-dried microencapsulation is an operationally simple method to protect active compounds from adverse reactions, harsh environments, and interactions with other ingredients. It improves the solubility of the active substance, increases stability in adverse environments, and masks unpleasant smells. Guo et al. [5] investigated the effect of spray drying and freeze drying on the microencapsulation of curcumin; freeze-dried encapsulated curcumin had a higher encapsulation rate, whereas the particles produced by spray drying had a smaller particle size, a smoother particle surface, and a more regular shape. Anthero et al. [6] emulsified and spray-dried microencapsulated paprika oleoresin using gum arabic and modified starches as wall materials to improve the solubility of the target. Ferraz et al. [7] prepared paprika and cinnamon oleoresin via the spray drying method, using whey protein isolate and maltodextrin as the wall materials, slowing down the rate of degradation of the active compounds under high temperatures and long-term storage conditions and reducing the pungency of the oils. Under certain conditions, encapsulation can achieve relatively high efficiency and stability, and effective controlled release.

The spray-drying microencapsulation preparation process is typically divided into two steps: (1) the encapsulated and microencapsulated core materials are dissolved or dispersed in water via auxiliary drive measures (e.g., stirring, homogenization, or ultrasonication) to prepare a primary emulsion or suspension; and (2) the emulsion or suspension is atomized and spray-dried to form the final microencapsulated product [8]. There are a variety of methods to promote the aqueous phase dissolution of fat-soluble active ingredients; currently, commonly used methods include stirring, high-speed shearing, colloidal milling, ultrasonic treatment, and high-pressure homogenization, often utilizing one or more of the above to improve the yield and encapsulation efficiency. Wang et al. [9] reported encapsulation rates of 19.9–97.9% for lutein microcapsules prepared with inulin and modified starch as encapsulating materials using high-speed shearing, wet media milling, high-pressure homogenization, and colloidal milling. Anthero et al. [10] tested stabilized capsicum oleoresin emulsions prepared by mixing capsicum oleoresin resin and aqueous phases with a mixer at a speed of 5000 rpm/min, pre-dissolving biopolymers of gum arabic, OSA-modified maize starch, and stearic acid-modified malt as biopolymers for 12 h in distilled water with magnetic stirring. The average droplet size of the emulsions was 1.66–18 μm. Liu et al. [11] homogenized an oil–water mixture at 10,000 r/min for 1 min and ultrasonicated it for 5 min (460 W, ≤45 °C) to obtain a capsaicin emulsion. There are a variety of drivers to assist in the wall encapsulation of targets at the laboratory stage, and practical applications and production require long-term validation.

Commonly used industrial techniques, including high-pressure homogenization and ultrasonic induction methods, are widely used for emulsion production in the food and pharmaceutical industries [12]. The use of these two methods for the preparation of paprika oleoresin microcapsules has been less studied. Therefore, in the current study, we utilized these two different techniques to prepare spray-dried microencapsulated paprika oleoresin powder. Subsequently, we evaluated the effects of the physicochemical properties, encapsulation rate, in vitro digestion, antioxidant capacity, and storage stability of the paprika oleoresin microcapsules.

## 2. Results and Discussion

### 2.1. Performance Parameters of Embedded Emulsions

The particle sizes and polydispersity index (PDI) of PO emulsions prepared via ultrasonication and high-pressure homogenization were determined, as shown in Figure 1A. The results showed that there was a difference in the particle size distributions of emulsions prepared in different ways, all of which were nano-sized emulsions, and the particle size became smaller with increases in strength. Overall, high-pressure homogenization produced smaller droplets than ultrasonic homogenization, while the H-100 particle size could be as small as 255 nm. PDI showed the same trend as particle size. The PDI of H-50 and H-100 emulsions was less than 0.2, which indicated uniform particle size. High-pressure homogenization reduces the size of nanoemulsions through mechanical forces such as high-speed shear, high-frequency oscillation, cavitation phenomena, and convective impact, while ultrasound reduces the size of nanoemulsions through cavitation [13]. As far as ultrasound-assisted methods are concerned, particle fragmentation can be achieved more efficiently using larger intensities. For high-pressure homogenization, higher intensities do not necessarily result in more complete particle breakage, and may result in wasted energy. The zeta potentials of the PO-loaded emulsions obtained in this study ranged between −28.1 mV and −35.1 mV (Figure 1B). The zeta potential of an emulsion system is usually determined by the type of encapsulated materials (EMs), and EMs can have a decisive effect on the pH, conductivity, and ionic strength of the emulsion [14]. However, the same EMs were used in this study; the reason for differences in the zeta potential value could be the particle size of the solution, as the higher the intensity of the mechanical energy impact, the more particles are produced and the smaller the particle size, which also confirms the results of Figure 1A. The desired zeta potential (typically potential modulus greater than 20) provides electrostatic repulsion and reduces the aggregation of paprika oleoresin particles during preparation [15]. The H-100 samples exhibited the lowest particle size and highest surface charge. The above results indicate that the size and potential of PO particles differed significantly among the five encapsulated emulsions.

Figure 2 shows before and after pictures obtained by leaving the emulsions at room temperature for a week. For freshly prepared emulsions, it was observed that the CK, U-200, and U-600 emulsions had a deeper reddish colour than the H-50 and H-100 emulsions. The reason may be that the first three samples were not well homogenized, and the particles between the oils were larger and aggregated with each other, so the emulsions showed darker colours. In contrast, after 7 d, the emulsions of the first three groups were lighter in colour than at 0 d, and a large amount of oil was uplifted. In contrast, the H-50 and H-100 emulsions did not change significantly in colour after 7 d, and less oil floated to the surface. The above results indicate that the samples of group H were more stable, which might have been related to their smaller particle size and larger potential.

### 2.2. Characterization of Spray-Dried Powders

Table 1 showed the performance parameters of the five PO microencapsulated powders in terms of encapsulation efficiency (EE), moisture content, solubility, and wettability. Encapsulation efficiency refers the ability of the microencapsulated carrier to encapsulate paprika oleoresin. The EEs of the five spray-dried powders ranged from 85.47% to 99%. The powders prepared via ultrasound and high-pressure homogenization were better embedded, both achieving more than 94.8%, with little difference in EE between the different groups. Of these, U-600 and H-100 had relatively high EE values (U-600: 99%; H-100: 98.6%). It was observed that the EE value increased with an increase in driving strength and a decrease in PO particle size in the emulsion. This phenomenon can be explained by the fact that when a certain amount of core composition is uniformly embedded and distributed in microencapsulated spheres, more core particles can be more completely protected.

In addition, as can be seen from Table 1, the moisture content of all five microencapsulated products was less than 3%, which was beneficial to the inhibition of microbial growth, and this moisture content was sufficient to ensure the storage stability of the PO microcapsules [16]. The different methods of preparing the emulsions did not significantly affect the resulting moisture content of the dried powders (*p* > 0.05).

One of the purposes of microencapsulation is to promote the uniform stabilization of fat-soluble substances in water. Therefore, good solubility is a key factor in promoting the application of microencapsulated products in the food industry [17]. It was observed that the solubility of all five microcapsules in water was more than 92.53%, as the two EMs belonged to the same carbohydrates and improved the overall solubility of the powders, and the solubility increased with an increase in driving strength under the same driving equipment conditions. U-600 microcapsules had the highest solubility (96.6%) and the largest EE value, while CK microcapsules without any treatment had the lowest solubility (92.53%) and the smallest EE value. This could have been the result of a higher surface oil content affecting the solubility of the microcapsules.

Wettability reflects the rehydration capacity of the microcapsules. From Table 1, it can be seen that the wettability of H-50 and H-100 was best, with wetting times of 278.04 s and 197.32 s, respectively. Wettability has been reported to be related to the size of the particles in the system [18]. The particle size of the microencapsulated product has a positive effect on the powder immersion time. The smaller the particle size, the larger the specific surface area, and the more easily it interacts with water, increasing wettability and shortening the time it takes for the microencapsulated powder to completely dissolve.

The colour value (CV) can reflect the pigment content to some extent. Observation of the colour value column showed that the CV values of U-600, H-50, and H-100 were significantly larger (U-600: 1685.89; H-50: 1703.58; H-100: 1649.68) (*p* < 0.05), indicating that these three powder products contained a greater amount of PO. The CV can be used to assess the pigment content and examine the PO residue of microencapsulated products.

### 2.3. Morphological Structure of Spray-Dried Powders

Some of the physicochemical properties of microcapsules are related to the morphology of the microencapsulated powder. Scanning electron microscope (SEM) images of microcapsules consisting of the encapsulated materials, inulin and sodium starch octenyl succinate are shown in Figure 3. As can be seen from the SEM images, the edges of the microcapsule particles were distinct and there was no adhesion. The external structure of the microcapsule particles was almost spherical or nearly spherical, with a heterogeneous particle size, and the spherical surface was wrinkled with a certain degree of surface depression, which is a common phenomenon in spray drying [19]. These particles also exhibited good structural integrity, with no visible pores or ruptures on the outer surface, suggesting that the PO was instantly encapsulated into the carbohydrate matrix with low oxygen permeability [20]. Although it has been demonstrated that EMs affect the surface morphology of spray-dried particles [21], the same EM was used in this study to evaluate the effect of the emulsion preparation method on the morphology of sprayed powders. With increasing driving intensity (stirring–ultrasound–high-pressure homogenization), the particle surface changed from a smooth spherical shape to shrinkage collapse. Under the same preparation conditions, the greater the induced intensity, the more pronounced the concave crumpling. This suggested that high-intensity mechanical forces enhanced the film-forming properties of the carbohydrates, which may have led to greater shrinkage and greater collapse.

### 2.4. Fourier Transform Infrared Spectroscopy (FTIR) Analysis of Paprika Oleoresin Microcapsules

FTIR can detect the introduction of new functional groups or the formation of new chemical bonds in PO microcapsules, and can characterize their molecular structure. Figure 4A shows the FTIR spectra of inulin, sodium starch octenyl succinate, paprika oleoresin, and the microcapsules. In this study, inulin and sodium starch octenyl succinate had similar FTIR spectra. Broad and strong stretching vibration peaks characteristic of –OH appeared in EMs at 3200–3600 cm^−1^. The stretching vibration peak of –CH was located at 2925 cm^−1^, the strong absorption peak of C=C was located at 1640 cm^−1^, and the strong absorption peak was located at 1025 cm^−1^. Moreover, the spectra of pure PO appeared near 3009 cm^−1^, 2925 cm^−1^, 2850 cm^−1^, 1740 cm^−1^, 1458 cm^−1^, and 722 cm^−1^, indicating the presence of aromatic rings, polar groups (hydroxyl, amide, and carbonyl), and hydrophobic groups [22]. In the spectral region 3000–3100 cm^−1^, the aromatic moiety was the most important band for this oleoresin. Meanwhile, the stretching vibrations of C–H groups in the regions of 2850 cm^−1^ and 722 cm^−1^ represented saturated aliphatic compounds [23].

It was found that the signal peak at 2850 cm^−1^ to 2925 cm^−1^ was associated with capsaicin, particularly paprika oleoresin stimulating compounds. The peaks at 1740 cm^−1^ and 1640 cm^−1^ correlated to the stretching vibration of C=O, and the peak at 1458 cm^−1^ corresponded to the stretching vibration of the N–H group. In addition, the band at 722 cm^−1^ characterized the regional vibrations of C–H and C–C, which were considered to be the aromatic bonds of the capsaicin benzene ring. The characteristic absorption peaks of the encapsulated paprika oleoresin powder corresponded to the EMs’ characteristic absorption peaks. Finally, loading chromatography of the particles showed band deformation at the carbonyl and hydroxyl groups, suggesting that PO may bind to the chemical groups of the encapsulated material.

### 2.5. Thermal Characterization of Paprika Oleoresin Microcapsules

The obtained differential scanning calorimetry (DSC) curves were analysed and compared using TA software (Version: 4.1.1.33073) to study the thermal properties of PO microcapsules, which was beneficial for evaluating product quality and stability. The DSC curves and glass transition temperatures (Tg) of the five PO microcapsules were shown in Figure 4B. Natural paprika oleoresin did not reach dissolution temperatures in the test range of 200 °C. The five PO microcapsules formed after spray drying showed a heat absorption peak near Tg 82 °C. This indicated that the glassy state of PO microcapsules could remain stable at room temperature. Therefore, the structure of the microcapsules should remain unchanged after heat treatment, which may be due to the stable structure of the PO microcapsules, containing starch, polysaccharides, and PO compounds. The Tg of the absorption peaks increased with increases in driving strength (Tg_H-100_ > Tg_H-50_; Tg_U-600_ > Tg_U-200_). Among them, U-600 had the highest absorption peak temperature and the highest EE (Table 1), and the maximum Tg of the product was 84.92 °C, which indicated the highest heat required for phase transition of the microcapsules and also implied that microcapsules prepared by ultrasound-induced emulsion had good thermal stability.

### 2.6. Digestive Capacity

PO release analysis was performed after 2 h of digestion in simulated intestinal fluid (SIF), and the results are shown in Figure 5A. Since EMs are carbohydrates, in vitro digestion mainly mimicked intestinal digestion. In SIF, the PO release from microcapsules prepared by ultrasound and high-pressure homogenization induction was significantly higher than that of the control group without any treatment (*p* < 0.05), while there was no significant difference in the release rate between microcapsules prepared by ultrasound and high-pressure homogenization (*p* > 0.05). The results showed that using ultrasound and high-pressure homogenization for the preparation of emulsions could improve the release characteristics of PO microcapsules and enhance bioavailability.

This may be because ultrasonication and high-pressure homogenization induced the formation of a more homogeneous solution, which facilitated the entry of PO into the cavity formed by the EMs and introduced more water-soluble hydroxyl groups to the particles, thus reducing the hydrophobicity of the PO microcapsules. Ultrasound and high-pressure homogenization induction improved the degradation of PO microcapsules in intestinal digestion by increasing the dispersibility and wettability of the PO microcapsule particles, thus accelerating the release of PO. Therefore, particles prepared by ultrasound and high-pressure homogenization induction showed better release characteristics in SIF.

### 2.7. Antioxidant Activities

The antioxidant capacity of the five types of microcapsules was investigated by analysing and comparing their free radical scavenging capacity. As shown in Figure 5B, the scavenging ability of U-200 microcapsules, U-600 microcapsules, H-50 microcapsules, and H-100 microcapsules for DPPH was 84.39%, 79.39%, 96.59%, and 89.19%, respectively, which was higher than that of CK particles (68.58%). This can be attributed to the fact that the microcapsules encapsulated hydrophobic PO or had surface hydrophilic EMs, which led to better dispersion of the particles in water and increased the contact between PO and free radicals. It was also possible that this was due to the degree of wrinkling on the particle surface: H-100 > H-50 > U-600 > U-200 > CK (Figure 3), which increased the contact area between PO and free radicals. In addition to this, H-50 and H-100 microcapsules showed greater free radical scavenging ability than U-200 and U-600 particles. In ABTS measurements, H-100 microcapsules exhibited the strongest scavenging ability, followed by H-50 particles (Figure 5B). The antioxidant activities determined by both methods demonstrated that the antioxidant capacity of microcapsules prepared by high-pressure homogenization was greater than that of microcapsules prepared by ultrasonication. The results indicated that using high-pressure homogenization induction to promote PO embedding and form PO microcapsules is an effective way to improve antioxidant capacity.

### 2.8. Stability against Adverse Conditions

The retention of PO in the five types of microcapsules was investigated under UV irradiation, as shown in Figure 6A. After 24 h of UV irradiation, U-600 particles had the highest PO retention, followed by H-100, H-50, and U-200 in that order, while CK microcapsules had the highest loss. The samples prepared by ultrasound and high-pressure homogenization had similar PO retention levels after short-term UV irradiation. This might have been due to the formation of hydrogen bonds between complex PO and EMs, and the presence of double bonds and aromatic residues in capsaicin [24]. In summary, PO encapsulated in composite microcapsules using ultrasonication and high-pressure homogenization showed greater stability under UV irradiation.

The thermal stability of PO is displayed in Figure 6B. After the five samples were held at 70 °C in a thermostat for 24 h, the PO retention rate of the particles did not differ much, except for the H-100 microcapsules, which had a lower PO retention rate. This may have been due to excessive mechanical strength that destroyed the physical barrier on the surface of the H-100 particles, allowing more exposure of the PO embedded in the cavities.

### 2.9. Influence of Light on Particle Stability

The effect of the preparation method on the stability of PO particles in terms of shape, colour, and PO content was evaluated by storing them for 30 days under two different treatments: natural light at room temperature and darkness at room temperature. Figure 7 demonstrates the change in the appearance of the particles over the storage period. The colour of all particles changed to some extent after storage, and there were various degrees of lightening of the colour of the particles.

Colour is an important parameter that influences consumers when purchasing food products. Carotenoids in PO were crucial in determining the colour of the particles [25]. The carotenoids present in chilli peppers all have chromogenic properties and can be classified into two different isochromatic fractions: red and yellow. The red fraction contains pigments specific to the capsicum genus (capsanthin, capsorubin, and capsanthin-5,6-epoxide). In contrast, the yellow fraction includes other pigments (zeaxanthin, violaxanthin, antheraxanthin, β-cryptoxanthin, β-carotene, and others), which are precursors of the first pigment [26]. The variation in carotenoid content in the microparticles determines the colouration state of the microcapsules.

Measurements of PO particles stored in different environments were evaluated using the CIELab system (a*, b*). Table 2 demonstrates the a* and b* of the particles stored for 30 days under natural light and darkness at room temperature. At the end of storage, all parameters of each formulation showed significant differences under the influence of sunlight (*p* < 0.05). The CK and U-200 particles exhibited a significant decrease in colour parameters (a* and b* values) under the influence of natural light (*p* < 0.05). Unexpectedly, U-600 particles exhibited a slight increase in colour parameters and less loss of carotenoids compared to the initial state, followed by H-50 and H-100 microcapsules, similar to the results in Figure 6A. The U-600 particles were better able to withstand an unfavourable environment during long-term storage, which may have been related to their higher peak temperature of thermal absorption (Figure 4B).

The results for storage stability of PO microcapsules in the presence or absence of natural light are shown in Figure 8, and the CV values were used to assess the total PO content in the microcapsules. After 30 d of exposure to natural light at room temperature (Figure 8A), the CV values of the five PO microcapsules (CK, U-200, U-600, H-50, and H-100) were 1003.65, 1153.45, 1163.34, 1155.73, and 1097.83, respectively. After 30 d of exposure to dark conditions at room temperature (Figure 8B), the CV values of the five microcapsules were 983.86, 1113.78, 1105.05, 1103.28, and 1059.99, respectively. The CV values of particles in groups U and H were higher than those of particles in the control group under either a natural light or dark environment, and the results indicated that PO microcapsules prepared by ultrasonication and high-pressure homogenization had good photoprotective effects on PO. In addition, the higher CV values of U-200, U-600, and H-50 microcapsules indicated that certain intensities of ultrasound and high-pressure homogenization induction effectively improved the photostability of PO.

## 3. Materials and Methods

### 3.1. Materials

PO was provided by Luck Huaguagn Graphics Co., Ltd. Sodium starch octenylsuccinate and inulin were purchased from Thousand Taste Food Technology Co., Ltd. (Shanghai, China), and Baiyun Aroma Chemical Co., Ltd. (Guangzhou, China), respectively. Organic reagents such as ethanol (99.7%) and hexane were of analytical grade and purchased from Shanghai Titan Technology Co., Ltd. (Shanghai, China). α-amylase was from Yuanye, Shanghai, China, trypsin, pancreatic lipase, and potassium persulfate from Aladdin, Shanghai, China, porcine bile salts from Solarbio, Beijing, China, and DPPH and ABTS were from Macklin, Shanghai, China.

### 3.2. Preparation of Embedded Emulsions

The process of preparing the emulsion as well as the microcapsules is shown in Figure 9. Sodium starch octenylsuccinate (29 g) and inulin (29 g) were dispersed as combined EMs in 140 g of water heated at 65 °C and stirred at 350 rpm/min for 30 min (Digital Display Constant Temperature Constant Speed Collecting Type Magnetic Heating Stirrer, DF-101SA, Changzhou Guoyu, Changzhou, China), then cooled to room temperature in cold water. A total of 2 g of paprika oleoresin was added dropwise to the aqueous phase prepared above, and then the mixture was stirred in a laboratory high-speed disperser (XHF-DY, Ningbo Xinzhi Biotechnology, Ningbo, China) at a speed of 5000 rpm for 5 min to form a homogeneous suspension, with a mass fraction of 30% of the suspension. The control group was abbreviated as CK.

Spray-dried emulsions of paprika oleoresin were separately prepared by ultrasound induction and high-pressure homogenization. (1) Ultrasonic induction method: the mixed solution was ultrasonicated for 10 min (pulse mode, 3 s on/3 s off) using a 6 mm probe ultrasonic cell crusher (JY99-IIDN, Ningbo Xinzhi Biotechnology, Ningbo, China) with output power of 200 W or 600 W, abbreviated as U-200 and U-600, respectively. (2) High-pressure homogenization method: the final emulsion was processed with an ultra-high-pressure homogenizer (207A, Ningbo Xinzhi Biotechnology, Ningbo, China) for 10 cycles at pressures of 50 MPa or 100 MPa. The treated samples were abbreviated as H-50 and H-100, respectively. The temperature of the emulsion was controlled at <35 °C during the preparation.

### 3.3. Spray Drying

PO particles were prepared using an experimental spray dryer (BILON-6000Y, Bilon Instruments, Shanghai, China). The PO suspension was continuously stirred during the spraying process and fed into the atomization chamber at a feed flow rate of 450 mL/h. The suction air rate and spray pressure were set at 0.55 m^3^/min and 0.5 MPa, respectively. The inlet temperature was set at 155 °C, and the outlet temperature varied between 75 and 85 °C. The spray-dried powder was collected from the bottom of the cyclone and transferred to a ziplock bag, and then stored at −18 °C.

### 3.4. Characterization of Emulsions

#### 3.4.1. Particle Size and Zeta Potential of Encapsulated Emulsions before Spray Drying

The particle sizes and polydispersity index (PDI) of the biopolymer dispersions and the zeta potential of the emulsions were measured using a Zetasizer Nano ZS (Malvern Instruments, Worcestershire, UK). The samples were diluted to 1 mg/mL with ultrapure water, and the measurements were repeated five times.

#### 3.4.2. Emulsion Stability

The five prepared emulsions were left in a room temperature environment for seven days to observe the suspension of the emulsions and to compare the colour, clarity, and floating of oils and fats via sensory comparison.

### 3.5. Characterization of Particles

#### 3.5.1. Moisture

Determination of moisture: the moisture of the particles was evaporated in an oven at 105 °C until a constant weight was reached. Then, the moisture content was determined based on the weight difference method.

#### 3.5.2. Embedding Efficiency

Determination of EE was based on the literature [27]. First, 15 mL of hexane was added to 1.5 g of powder and vortexed for 2 min. The mixture was filtered through filter paper and the collected powder was rinsed twice with 15 mL of hexane. The collected filtrate was placed in an oven at 60 °C for solvent evaporation until a constant weight was reached. The EE of PO was calculated using the weight difference method, according to the following equation:(1)$$\mathrm{EE}\;\left(\%\right)=\frac{\left(\mathrm{Total}\;\mathrm{oil}\;\mathrm{content}-\mathrm{surface}\;\mathrm{oil}\;\mathrm{content}\right)}{\mathrm{Total}\;\mathrm{oil}\;\mathrm{content}}\times 100$$

#### 3.5.3. Solubility Measurement

Pellets (0.2 g dry basis) were added to 20 mL of distilled water and stirred for 5 min. The pellets were centrifuged at 10,000× *g* rpm/min for 10 min at 20 °C using a centrifuge (H1650-W, Xiangyi, Hunan, China). A total of 5 mL of the supernatant was transferred to a pre-weighed petri dish and then dried in an oven at 105 °C for 5 h. Solubility was calculated based on the weight difference.
(2)$$\mathrm{Solubility}\;\left(\%\right)=\frac{\mathrm{Weight}\;\mathrm{of}\;\mathrm{supernatant}\;\mathrm{after}\;\mathrm{drying}\times 4}{\mathrm{Total}\;\mathrm{weight}}\times 100$$

#### 3.5.4. Wettability Measurement

An accurately weighed amount of microencapsulated powder (0.1 g) was sprinkled into 50 mL of deionized water at room temperature and then stirred with a magnetic stirrer at 450 rpm/min until completely submerged. We recorded the time at which the powder was fully immersed [28].

#### 3.5.5. Colours

To determine the colour of the particles, the powders were evaluated using a Colour Reader (CR9, 3nh, Guangzhou, China), D65 light source, and 10° angle, using the colour parameters (a*, b*) of the CIELab scale. a* represented the change in colour between red (+) and green (−), b* indicated the change in colour between yellow (+) and blue (−), and the a* and b* values were used as colour parameters after the samples were stored for 30 days under light and dark room temperature storage conditions.

#### 3.5.6. Colour Values

The powder obtained after spray drying was used to determine the CV according to the method of the American Spice Trade Association [29]. A certain amount of sample was accurately weighed, added to 4 mL of deionized water for ultrasonic treatment, and then added to 36 mL of anhydrous ethanol and centrifuged at 6000 r/min for 10 min at 4 °C to obtain the supernatant. The absorbance was measured at 460 nm using a UV–Vis spectrophotometer (UV1901PC, Shanghai Aoan Scientific Instruments, Shanghai, China) in a l cm cuvette, with 90% ethanol as the reference solution. We calculated the CV according to the following equation:(3)$$\mathrm{CV}=\frac{A\times f\times 166.7}{m\times 100}$$
where A is the absorbance of the samples after dilution, f is the dilution ratio, and m is the weight of the samples.

### 3.6. Characterization of Performance Features

#### 3.6.1. Thermal Analysis

Following the method of Ding et al. [30], the thermal properties of PO microcapsules were analysed using a differential scanning calorimeter (DSC3, Mettler Toledo, Nänikon, Switzerland) under a nitrogen atmosphere at a flow rate of 40 mL/min. Samples were sealed in aluminium crucibles and analysed in the range of 30–200 °C at a heating rate of 10 °C/min to obtain DSC curves. The DSC curves were processed and analysed using TA software.

#### 3.6.2. Morphologic Observation of Microcapsules

The morphology of the spray-dried powder samples was observed using a scanning electron microscope (GeminiSEM 300, ZEISS, Jena, Germany) at 10 kV. To obtain images (magnification 2000× and 15,000×), dried samples were taken and directly glued to a conductive adhesive sprayed with gold (SC7620, Quorum, East Sussex, UK), observed, and photographed.

#### 3.6.3. FTIR Analysis of Paprika Oleoresin Microcapsules

A Fourier transform infrared spectrometer (Niolet iN10, Thermo Fisher Scientific, Waltham, MA, USA) was utilized for determination and analysis. The samples and dried KBr powder were mixed, ground, and pressed into flakes. Scanning at 4 cm^−1^ on a background of KBr slices, the test wave numbers ranged from 400 to 4000 cm^−1^ [31].

### 3.7. In Vitro Digestion

A static in vitro intestinal digestion model was used to determine and analyse the bioavailability of PO in microcapsules obtained by spray drying. Simulated in vitro intestinal digestion was performed as described in the literature [32]. To prepare SIF, 2 g of sample was dispersed in a certain volume of SIF; trypsin, lipase, α-amylase, and porcine bile salt were added, and the pH of the mixture was adjusted to 7 to obtain 50 mg/mL of intestinal premix. The intestinal collage was placed in a 37 °C thermostat to be digested by oscillation (100 rpm) for 2 h. After removal, the samples were put into ice water for 30 min to stop the digestion. The digested mixture was centrifuged at 10,000× *g* for 15 min to obtain the supernatant.

The digested supernatant was collected. Absorbance was measured at 460 nm using a UV–Vis spectrophotometer, and CV was calculated using the equation in Section 3.5.6. The rate of PO release from the powder was expressed as the ratio of the PO content in the supernatant to the initial PO content.

### 3.8. In Vitro Antioxidant Capacity

#### 3.8.1. DPPH Free Radical Scavenging Assay

The 1,1-diphenyl-2-picrylhydrazyl (DPPH) radical scavenging capacity of the samples was determined according to a minor modification of the method described by Meng et al. [33]. Two millilitres of each PO sample (30 mg/mL) was added to 2.0 mL of ethanol solution of DPPH (0.16 mM), and mixed thoroughly. The mixed solution was left at 20 °C for 30 min, protected from light, and absorbance (D_1_) was measured at 517 nm. The absorbances of blank (D_2_) and control (D_0_) samples were determined using the same method. In the blank, ethanol was used instead of the DPPH solution; in the control, ethanol was used instead of sample solution. The DPPH free radical scavenging capacity was calculated using the equation:(4)$$\mathrm{DPPH}\;\mathrm{radical}\;\mathrm{scavenging}\;\mathrm{capacity}\;\left(\%\right)=(1-\frac{D_{1}-D_{2}}{D_{0}})\times 100$$

#### 3.8.2. ABTS Free Radical Scavenging Assay

The ABTS free radical scavenging capacity was assessed, with some modifications, according to Meng et al. [33]. ABTS (7 mM) was reacted with potassium persulfate (5 mM) at a 1:1 (*v*/*v*) ratio to generate ABTS radicals. The mixture was then placed in the dark at 25 °C for 16 h. The absorbance measured at 734 nm was 0.7 after dilution with PBS (10 mM, pH 7.4) to obtain the ABTS working solution. Then, 0.6 mL of sample (30 mg/mL), 0.6 mL of ABTS working solution, and 2.8 mL of PBS were mixed, reacted for 5 min, and absorbance (A_s_) was measured at 734 nm. The control (A_c_) used 10 mM PBS instead of the sample solution. We used the following formula to determine the removal capacity of the samples:(5)$$\mathrm{ABTS}\;\mathrm{scavenging}\;\mathrm{capacity}\;\left(\%\right)=\frac{A_{c}-A_{s}}{A_{c}}\times 100$$

### 3.9. Stability against Adverse Conditions

#### 3.9.1. Photochemical Stability of Paprika Oleoresin Microcapsules

The photochemical stability of PO microcapsules was determined following ultrasound and high-pressure homogenization treatments [34]. The five types of freshly prepared microcapsules were spread thinly on a glass dish and exposed to 254 nm UV light (8 W). The experimental sampling times were 0 h, 4 h, 8 h, 12 h, 16 h, and 24 h, and the colour values were determined according to the method in Section 3.5.6 to calculate the PO retention rate.

#### 3.9.2. Thermal Stability of Paprika Oleoresin Microcapsules

The samples were packed into transparent glass vials and kept warm in a thermostat at 70 °C. Sampling times were 0 h, 4 h, 8 h, 12 h, 16 h, and 24 h. PO retention was determined as above.

### 3.10. Storage Stability of Paprika Oleoresin Microcapsules

According to Zhang et al. [35], the effect of light on the chemical stability of microcapsules after different treatments was evaluated following long-term storage to assess the residual amount of PO. The five types of PO microcapsules were inserted into transparent glass vials and stored at room temperature under natural light for 30 days, while the control group was stored at room temperature and protected from light by covering with tin foil; 100 mg of the samples was taken every four days to determine the total lipid content and to calculate the retention rate (R) of PO in the microcapsules. The following formula was used to calculate the retention rate:(6)$$R\%=100\times \frac{\mathrm{Residual}\;\mathrm{paprika}\;\mathrm{oleoresin}\;\mathrm{content}}{\mathrm{Initial}\;\mathrm{paprika}\;\mathrm{oleoresin}\;\mathrm{content}}$$

### 3.11. Statistical Analysis

Data were presented as means with standard deviations. One-way ANOVA was performed using Prism 8 (GraphPad, La Jolla, CA, USA) and *p* < 0.05 was considered significant. Graphs were created using Prism 8 and Origin 2018 software (OriginLab Corporation, Northampton, PA, USA).

## 4. Conclusions

In this study, oil-in-water microcapsules were prepared by the spray drying method using paprika oleoresin as the core substance and inulin and sodium starch octenylsuccinate as the wall materials. The objective of this study was to improve the stability and water solubility of the active ingredients in the microencapsulated core material by induction of ultrasound and high-pressure homogenization. The results showed that the particles in the PO emulsion were nanoscale, and after the emulsion was left to stand for one week, PO in the emulsion prepared by high-pressure homogenization was less likely to float to the surface, and the emulsion was stable. The five types of microcapsules were structurally intact, with no obvious pores or ruptures on the outer surface, with smooth surfaces, while concave wrinkles were more obvious with increases in mechanical shear strength. Rehydration was continuously enhanced. The EE of the U-600 and H-100 microcapsule systems, with high mechanical strength, increased solubility, and the highest heat absorption capacity, showed good thermal and photochemical stability. Microcapsules produced by ultrasound and high-pressure homogenization preparations had comparable intestinal digestibility, which was significantly higher than that of the control, and microcapsules prepared by high-pressure homogenization showed significant antioxidant activity. The PO microcapsules produced by 600 W ultrasonication showed a positive effect on the retention of lipid content in both short-term UV irradiation and long-term natural light storage tests.

These results indicate that the spray drying method is effective for the preparation of PO using ultrasound and high-pressure homogenization as induction means. PO microcapsules are characterized by uniformity and stability, high digestibility, good water solubility, and high content, and they can be added to feeds as dietary supplements to improve the absorption of carotenoids in poultry. Our findings provide a basis for the use of ultrasonic and high-pressure homogenization equipment for microcapsule production to prepare emulsions containing PO microcapsules, prolonging the shelf-life of this bioactive product and improving its oral bioavailability.

## Figures and Tables

**Figure 1 molecules-28-07075-f001:**
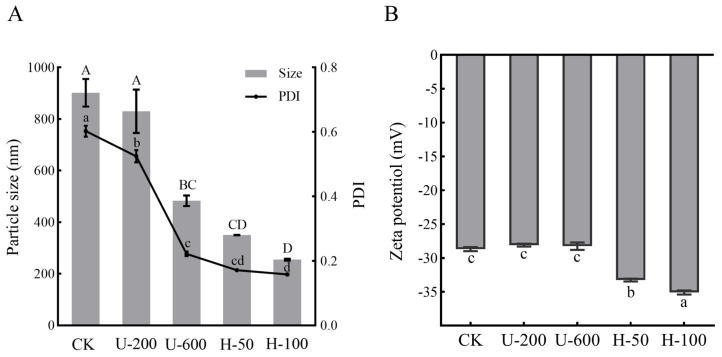
Particle size and polydispersity index (PDI) (**A**) and zeta potential (**B**) of five encapsulated emulsions prepared through ultrasound—induced and high—pressure homogenization methods (different letters in the graph indicate significant differences (*p* < 0.05); mean ± SD, *n* = 5).

**Figure 2 molecules-28-07075-f002:**
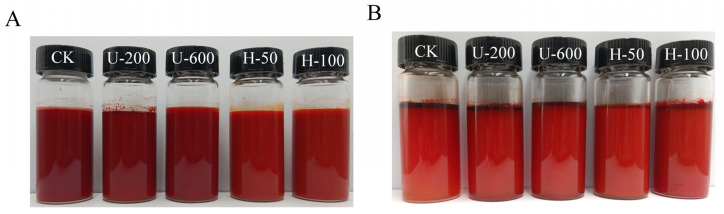
Emulsions kept at room temperature for seven days ((**A**): 0 d; (**B**): 7 d).

**Figure 3 molecules-28-07075-f003:**
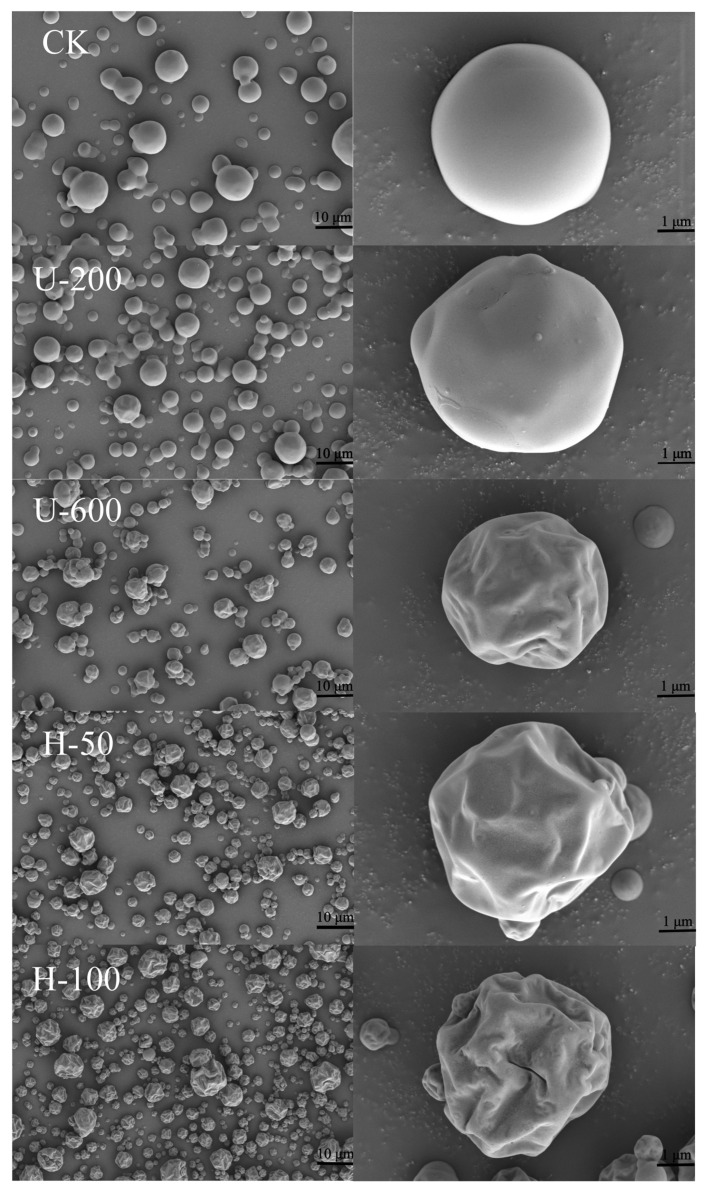
Scanning electron microscope (SEM) images of paprika oleoresin particles produced with different preparations.

**Figure 4 molecules-28-07075-f004:**
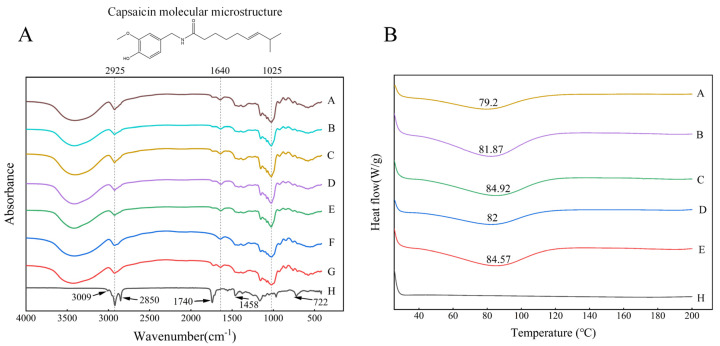
(**A**) Fourier transform infrared spectroscopy (FTIR) and (**B**) differential scanning calorimetry (DSC) plots of inulin–sodium starch octenylsuccinate–paprika oleoresin microcapsules. (A: CK; B: U-200; C: U-600; D: H-50; E: H-100; F: inulin; G: sodium starch octenylsuccinate; H: paprika oleoresin).

**Figure 5 molecules-28-07075-f005:**
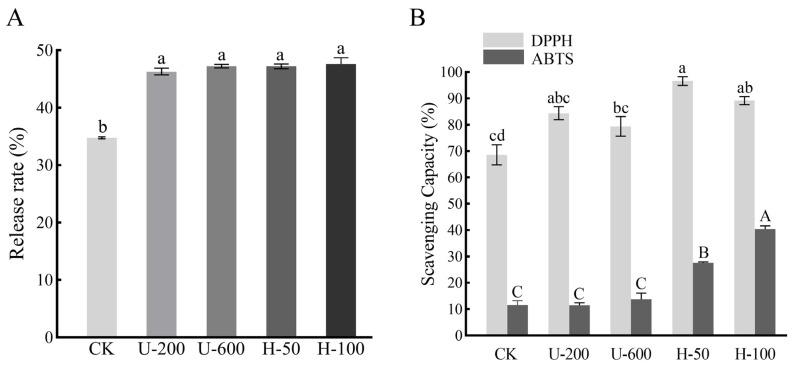
Release profile (**A**) and DPPH and ABTS radical scavenging ability (**B**) of CK, U-200, U-600, H-50, and H-100. Different letters indicate significant differences in the results (*p* < 0.05).

**Figure 6 molecules-28-07075-f006:**
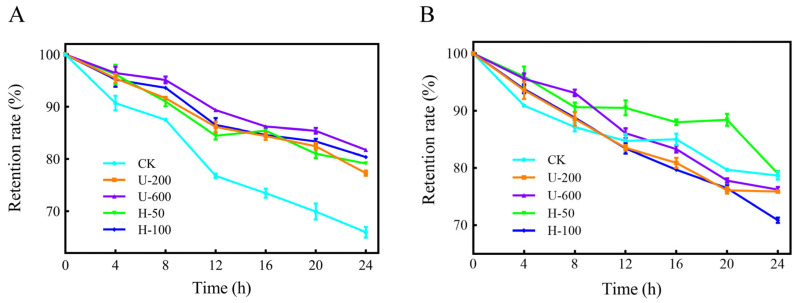
Light stability (**A**) and thermal stability (**B**) of CK, U-200, U-600, H-50, and H-100 microcapsules.

**Figure 7 molecules-28-07075-f007:**
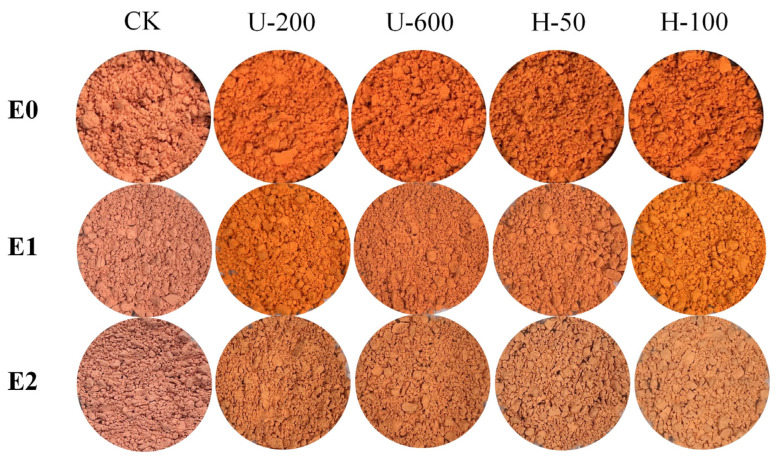
Visual appearance of paprika oleoresin microparticles over 30 days of storage at room temperature (sunlight or darkness). E0: initial state at 0 days; E1: sunlight at room temperature; E2: darkness at room temperature.

**Figure 8 molecules-28-07075-f008:**
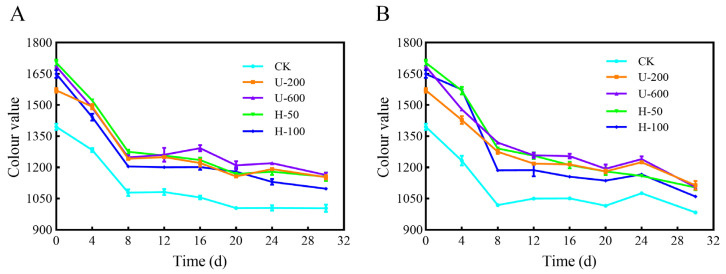
Colour value of paprika oleoresin over 30 days of storage at room temperature (sunlight or darkness). (**A**) Sunlight at room temperature; (**B**) darkness at room temperature.

**Figure 9 molecules-28-07075-f009:**
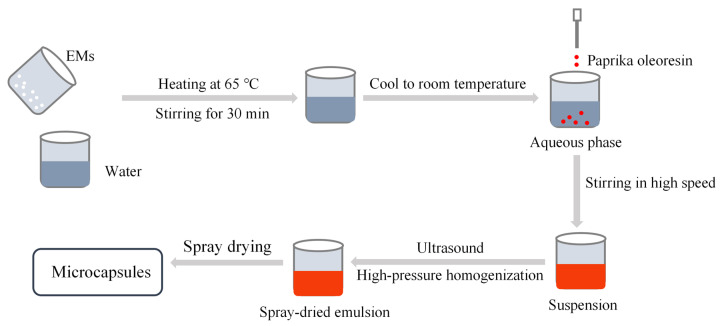
Schematic diagram of the induced preparation of paprika oleoresin microcapsules by ultrasonication and high-pressure homogenization.

**Table 1 molecules-28-07075-t001:** Performance parameters of paprika oleoresin microencapsulated powders prepared using different methods.

Sample	EE (%)	Moisture (%)	Solubility (%)	Wettability (s)	CV
CK	85.47 ± 1.43 ^e^	2.81 ± 0.31 ^a^	92.53 ± 0.82 ^b^	456.83 ± 6.14 ^a^	1395.18 ± 21.22 ^e^
U-200	97.40 ± 0.75 ^ca^	2.73 ± 0.04 ^a^	95.47 ± 1.23 ^ab^	352.25 ± 4.82 ^b^	1570.59 ± 14.7 ^d^
U-600	99.00 ± 0.33 ^a^	2.57 ± 0.19 ^a^	96.60 ± 1.45 ^a^	330.51 ± 1.53 ^c^	1685.89 ± 10.3 ^ba^
H-50	94.80 ± 0.33 ^d^	2.33 ± 0.29 ^a^	94.47 ± 1.43 ^ab^	278.04 ± 0.95 ^d^	1703.58 ± 20.58 ^a^
H-100	98.60 ± 0.16 ^ba^	2.54 ± 0.30 ^a^	96.20 ± 1.02 ^ab^	197.32 ± 2.24 ^e^	1649.68 ± 26.73 ^cba^

All values are presented as mean ± SD (*n* = 3). ^a–e^ Different letters within each column indicate significant differences (*p* < 0.05). EE: encapsulation efficiency; CV: colour value.

**Table 2 molecules-28-07075-t002:** Colour parameters of paprika oleoresin microcapsules stored for 30 days in light at room temperature and darkness at room temperature.

Days	Samples	a*	b*	Days	Samples	a*	b*
	CK				U-200		
0	E0	30.22 ± 0.22 ^a^	26.16 ± 0.1 ^a^	0	E0	41.7 ± 0.07 ^a^	51.77 ± 0.22 ^a^
30	E1	28.31 ± 0.08 ^b^	25.2 ± 0.11 ^b^	30	E1	40.03 ± 0.33 ^b^	48.62 ± 0.41 ^b^
	E2	27.72 ± 0.15 ^c^	25.41 ± 0.13 ^b^		E2	39.46 ± 0.04 ^b^	48.4 ± 0.07 ^b^
	U-600				H-50		
0	E0	36.58 ± 0.32 ^b^	47.43 ± 0.53 ^a^	0	E0	37.7 ± 0.12 ^b^	48.51 ± 0.39 ^a^
30	E1	39.49 ± 0.38 ^a^	47.35 ± 0.61 ^a^	30	E1	38.29 ± 0.26 ^a^	48.06 ± 0.36 ^a^
	E2	38.64 ± 0.07 ^a^	48.24 ± 0.21 ^a^		E2	37.27 ± 0.16 ^b^	47.72 ± 0.07 ^a^
	H-100				H-100		
0	E0	38.98 ± 0.13 ^a^	56.34 ± 0.24 ^a^	30	E2	38.8 ± 0.06 ^a^	54.33 ± 0.11 ^b^
30	E1	39.03 ± 0.08 ^a^	53.28 ± 0.09 ^c^				

E0: Initial state at 0 days; E1: sunlight at room temperature; E2: darkness at room temperature; a*: Chromaticity (+a* = red and −a* = green); b*: Chromaticity (+b* = yellow and −b* = blue); mean ± standard deviation; averages with different lowercase letters in the same row show a significant difference (*p* < 0.05) using Tukey’s test for colour parameters between storage periods.

## Data Availability

Not applicable.

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
