# Peer review of "Characterization of Spray-Dried Microcapsules of Paprika Oleoresin Induced by Ultrasound and High-Pressure Homogenization: Physicochemical Properties and Storage Stability"

_molecules, 2023, doi:10.3390/molecules28207075_

Round 1
Reviewer 1 Report
The manuscript deals with an interesting topic. However, please consider the following points
Introduction:
Lines 37-41: What is the relationship between high viscosity and solubility with PO degradation?
Materials and methods
Line 355: insert the equipment and stirring speed used
Line 363: Emulsion is a dispersed system. Was the term solution used correctly in this sentence?
Line 387: “ the suspension of the emulsion”. What would be emulsion suspension? describe in more detail the physicochemical and organoleptic parameters evaluated in the study.
Section 3.5.3. insert equation used to calculate solubility
Line 419: “ samples were stabilized” Wouldn't the correct term be "stored"?
I believe that a careful review of terms used throughout the manuscript, solution/dispersion, nanospheres/microspheres, among others, is necessary so that the terms are standardized.
For example: line 495 “ PO nanoparticles” and line 538 “PO microcapsules”.
Line 508: “ Po microcapsules were encapsulated in transparente glass vials” Would encapsulated be the correct term? inserted, stored would not be more appropriate?
Results and discution
Line 103-104: the author refers to the zeta potential of emulsions with values ​​between 23 and 28 mV and cites figure 1B. Check the values ​​mentioned with those shown in the graph.
Line 106-108: the author used the term zeta potential less than - 20 mv. Wouldn't it be more interesting to use greater than 20 in modulus so as not to have wrong interpretations? Since the negative sign refers to the charge of the particle and the zeta potential value needs to be greater than 20 mV to have greater stability.
“The above results indicated that the size and potential of 109 PO particles were the main differences between the five encapsulated emulsions.” Rewrite sentence since only zeta potential and particle size have been evaluated so far.
It would be interesting to explain how the obtaining method influences the zeta potential. Why does homogenization increase the zeta potential value?
Explain why the emulsions show color variation depending on the method of production when evaluated on day 0.
Explain why they all showed the same color tone after 7 days of storage, a similar tone to the H-50 and H-100 emulsions on day 0.
Line 146 “solubility increased with the increase of driving strength”. Why then did the H100 formulation not have the highest solubility?
Line 165: “indicating that 164 these three powder products contained a greater amount of PO” in relation to what? free and encapsulated? Wasn't the same amount used in all emulsions?
Line 173: “uneven in size” heterogeneous particle size?
Author Response
For research article
|
Response to Reviewer 1 Comments
|
||
|
1. Summary |
|
|
|
We are very grateful to you for your professional and constructive suggestions on our manuscript (molecules-2632060). These suggestions have been valuable and helpful in improving our paper. We have carefully revised the manuscript to your and the reviewers' requests and suggestions. Responses to the reviewers' comments are also provided, and this information will be fully presented in the next few pages.
|
||
|
2. Questions for General Evaluation |
Reviewer’s Evaluation |
Response and Revisions |
|
Does the introduction provide sufficient background and include all relevant references? |
Yes |
|
|
Are all the cited references relevant to the research? |
Yes |
|
|
Is the research design appropriate? |
Yes |
|
|
Are the methods adequately described? |
Can be improved |
We can give your corresponding response in the point-by-point response letter. |
|
Are the results clearly presented? |
Must be improved |
We can give your corresponding response in the revised manuscript. |
|
Are the conclusions supported by the results? |
Can be improved |
We can give your corresponding response in the point-by-point response letter. |
|
3. Point-by-point response to Comments and Suggestions for Authors |
||
|
Comments: The manuscript deals with an interesting topic. However, please consider the following points. Response: Thank you very much for your positive comments on this manuscript, and we will carefully revise it according to your suggestions to improve its quality.
Introduction Comments 1: Lines 37-41: What is the relationship between high viscosity and solubility with PO degradation? |
||
|
Response 1: Thank you very much for your professional question. Higher viscosity indicates higher PO content, which slows down the rate of PO degradation. The water solubility of PO is low because it is a fat, and encapsulation of PO in the form of microcapsules and other forms improves its solubility, while hopefully slowing down the degradation of PO and making it more absorbed and utilised by the organism. |
||
|
Materials and methods Comments 2: Line 355: insert the equipment and stirring speed used Response 2: Thank you for your kind reminder, we have added the device information in the appropriate place in the manuscript. (Line 365-367)
Comments 3: Line 363: Emulsion is a dispersed system. Was the term solution used correctly in this sentence? Response 3: Thanks for your kind reminder. Emulsion is appropriate for this situation. In this study, oil-in-water emulsions of sodium starch octenylsuccinate, inulin, and paprika oleoresin were prepared to form nanoscale emulsions.
Comments 4: Line 387: “the suspension of the emulsion”. What would be emulsion suspension? describe in more detail the physicochemical and organoleptic parameters evaluated in the study. Response 4: We appreciate your suggestions and the organoleptic parameters are described in detail in "3.4.2 Emulsion Stability".
Comments 5: Section 3.5.3. insert equation used to calculate solubility Response 5: Thank you for your kind reminder, we have inserted the equation in the corresponding place in the manuscript.
Comments 6: Line 419: “samples were stabilized” Wouldn't the correct term be "stored"? Response 6: Thank you very much for spotting the problem in this section, we have changed the content of the original manuscript to " stored". (Line 437)
Comments 7: I believe that a careful review of terms used throughout the manuscript, solution/dispersion, nanospheres/microspheres, among others, is necessary so that the terms are standardized. For example: line 495 “PO nanoparticles” and line 538 “PO microcapsules”. Response 7: Thanks to your kind reminder, we have made corrections in the manuscript in the corresponding places (lines 266, 267, 269, 273, 275, 280, 282, 284, 289, 293, 298, 326, 475, 512, 565).
Comments 8: Line 508: “Po microcapsules were encapsulated in transparent glass vials” Would encapsulated be the correct term? inserted, stored would not be more appropriate? Response 8: The authors greatly appreciate your professional advice. We have changed the “encapsulated “in the original manuscript to “inserted “. (Line 525)
Results and discussion Comments 9: Line 103-104: the author refers to the zeta potential of emulsions with values ​​between 23 and 28 mV and cites figure 1B. Check the values ​​mentioned with those shown in the graph. Response 9: Thank you very much for spotting the problem in this section. After checking the values shown in the figure, we have changed the values in the original manuscript.
Comments 10: Line 106-108: the author used the term zeta potential less than -20 mv. Wouldn't it be more interesting to use greater than 20 in modulus so as not to have wrong interpretations? Since the negative sign refers to the charge of the particle and the zeta potential value needs to be greater than 20 mV to have greater stability. Response 10: The authors greatly appreciate your professional advice. We have changed "typically less than -20 mV" in the original manuscript to " typically potential modulus greater than 20". (Line 111-112)
Comments 11: “The above results indicated that the size and potential of PO particles were the main differences between the five encapsulated emulsions.” Rewrite sentence since only zeta potential and particle size have been evaluated so far. Response 11: Thank you very much for your suggestion, we have amended the original sentence to “The above results indicated that the size and potential of PO particles differed significantly among the five encapsulated emulsions”. (Lines 114-116)
Comments 12: It would be interesting to explain how the obtaining method influences the zeta potential. Why does homogenization increase the zeta potential value? Response 12: Thank you for your reminder and suggestions. Influencing the zeta potential value could be the particle size of the solution in addition to the EMs; the higher the intensity of the mechanical energy of the impact, the more particles are produced and the smaller the particle size, resulting in a larger potential value. The relevant reasons have been repeated in the original text. (Line 108-111)
Comments 13: Explain why the emulsions show color variation depending on the method of production when evaluated on day 0. Response 13: We greatly appreciate your suggestion. The reason may be that the samples of CK, U-200, and U-600 were not well homogenised, and the particles of oil and grease were large and aggregated with each other, so the emulsion showed a darker colour. (Line 123-126)
Comments 14: Explain why they all showed the same color tone after 7 days of storage, a similar tone to the H-50 and H-100 emulsions on day 0. Response 14: Thank you very much for your professional question. The emulsion was left at room temperature for 7d, the grease on the surface of the microcapsules or the grease that was not tightly encapsulated, floated up by its own density, resulting in a decrease in the amount of grease present in the emulsion, so that the emulsion showed a similar tone. The colour tone of the emulsion at 7d was similar to that of the H-50 and H-100 emulsions at 0d, probably due to the uniform distribution of the oil in the emulsion and the solid binding of the oil to the encapsulating materials.
Comments 15: Line 146 “solubility increased with the increase of driving strength”. Why then did the H-100 formulation not have the highest solubility? Response 15: Thanks for your suggestion. We have corrected Line 146 “solubility increased with the increase of driving strength” to Line 152 “the solubility increased with the increase of driving strength under the same driving equipment conditions” in the original text. (Line 154-155) In Table 1, the solubility of the two formulations with higher solubility, U-600 and H-100, were comparable and there was no significant difference. The solubility of the H-100 formulation was not the highest, probably because the solubility increased with the increase of driving strength within a certain range, and too much driving strength might disrupt the bonding of the grease with the encapsulating material, affecting the solubility of the microcapsules.
Comments 16: Line 165: “indicating that these three powder products contained a greater amount of PO” in relation to what? free and encapsulated? Wasn't the same amount used in all emulsions? Response 16: Thank you for your professional questions regarding the content of our study. The level of PO content encapsulated in the powder product was assessed by CV values, and using the same amount of PO in this study, the test results found that U-600, H-50, and H-100 had larger CV values, indicating that more PO was encapsulated and that these driving methods were feasible.
Comments 17: Line 173: “uneven in size” heterogeneous particle size? Response 17: The author appreciates your suggestions and has made the changes. We have edited "uneven in size" in the original manuscript to "heterogeneous particle size". (Line 181) |
||
Reviewer 2 Report
The manuscript presented by Zhang et al. seems interesting and has the potential to be published in Molecules after considering some minor revisions:
1- I also recommend report of PDI in the section where hydrodynamic diameters and surface charge are analyzed.
2- SEM pictures should present a clear scale bar for better understanding.
3- Section 3.2. I suggest incorporating a scheme for better visualization and comparison between ultrasound induction and high-pressure homogenization procedures.
4- Equations should be numbered.
Best,
Author Response
For review article
|
Response to Reviewer 2 Comments
|
||
|
1. Summary |
|
|
|
Thank you very much for taking the time to review this manuscript (molecules-2632060). Responses to the reviewers' comments are provided, please find the detailed responses below and the corresponding revisions in the resubmitted files.
|
||
|
2. Questions for General Evaluation |
Reviewer’s Evaluation |
Response and Revisions |
|
Does the introduction provide sufficient background and include all relevant references? |
Yes |
|
|
Are all the cited references relevant to the research? |
Yes |
|
|
Is the research design appropriate? |
Yes |
|
|
Are the methods adequately described? |
Can be improved |
We can give your corresponding response in the point-by-point response letter. |
|
Are the results clearly presented? |
Can be improved |
We can give your corresponding response in the point-by-point response letter. |
|
Are the conclusions supported by the results?
|
Yes |
|
|
3. Point-by-point response to Comments and Suggestions for Authors |
||
|
Comments: The manuscript presented by Zhang et al. seems interesting and has the potential to be published in Molecules after considering some minor revisions: Response: Thank you very much for your support and comments on this manuscript. We will carefully revise it based on your comments and those of other reviewers and hope to publish it in the Molecules journal.
Comments 1: I also recommend report of PDI in the section where hydrodynamic diameters and surface charge are analyzed. |
||
|
Response 1: Thanks very much for your kind reminders and suggestions. We have added PDI reports to the emulsion particle size and charge sections by adding the PDI data in Figure 1A. (Lines 90, 96-97, 395)
|
||
|
Comments 2: SEM pictures should present a clear scale bar for better understanding. |
||
|
Response 2: Thank you for the kind reminder, we have processed the SEM images to show a clear scale bar.
Comments 3: Section 3.2. I suggest incorporating a scheme for better visualization and comparison between ultrasound induction and high-pressure homogenization procedures. Response 3: We appreciate your suggestions and the related schematic is shown in the manuscript. (Line 363, 383-384)
Comments 4: Equations should be numbered. Response 4: The authors greatly appreciate your professional advice and we have numbered the equations in the text. |
||